# Performance of Commercial Dermatoscopic Systems That Incorporate Artificial Intelligence for the Identification of Melanoma in General Practice: A Systematic Review

**DOI:** 10.3390/cancers16071443

**Published:** 2024-04-08

**Authors:** Ian Miller, Nedeljka Rosic, Michael Stapelberg, Jeremy Hudson, Paul Coxon, James Furness, Joe Walsh, Mike Climstein

**Affiliations:** 1Aquatic Based Research, Southern Cross University, Bilinga, QLD 4225, Australia; i.miller.11@student.scu.edu.au (I.M.); nedeljka.rosic@scu.edu.au (N.R.);; 2Faculty of Health, Southern Cross University, Bilinga, QLD 4225, Australiapmcoxon@gmail.com (P.C.); 3Specialist Suite, John Flynn Hospital, Tugun, QLD 4224, Australia; 4North Queensland Skin Centre, Townsville, QLD 4810, Australia; 5Water Based Research Unit, Bond University, Robina, QLD 4226, Australia; jfurness@bond.edu.au; 6Sport Science Institute, Sydney, NSW 2000, Australia; research@sportscienceinstitute.com; 7AI Consulting Group, Sydney, NSW 2000, Australia; 8Physical Activity, Lifestyle, Ageing and Wellbeing Faculty Research Group, University of Sydney, Sydney, NSW 2050, Australia

**Keywords:** computer-aided diagnosis, convolutional neural network, deep learning, dermatology, detection, diagnosis, epidemiology, machine learning, skin cancer, total body photography

## Abstract

**Simple Summary:**

Early detection of malignant melanoma (MM) has the potential to significantly reduce morbidity and mortality as the thickness of the lesion is closely associated with prognosis. Artificial intelligence (AI) is a non-invasive technology that has the potential to aid clinicians in the early diagnosis of skin cancers, including melanoma. Performance metrics of machine-based AI have been shown to rival or improve upon clinician diagnosis. However, many studies have limited this scope to pre-build image databases that may not replicate real-world settings experienced in general practice. This systematic review aimed to report the performance of commercial or market-approved dermatoscopic systems with AI in the form of convolutional neural networks (CNNs) when tasked with classifying melanoma. The sensitivity and specificity of CNNs are highly varied and illustrate the necessity of clinician-to-patient interaction in the diagnosis process. Clinicians working in unison with AI show the most promise for better performance when classifying MM.

**Abstract:**

Background: Cutaneous melanoma remains an increasing global public health burden, particularly in fair-skinned populations. Advancing technologies, particularly artificial intelligence (AI), may provide an additional tool for clinicians to help detect malignancies with a more accurate success rate. This systematic review aimed to report the performance metrics of commercially available convolutional neural networks (CNNs) tasked with detecting MM. Methods: A systematic literature search was performed using CINAHL, Medline, Scopus, ScienceDirect and Web of Science databases. Results: A total of 16 articles reporting MM were included in this review. The combined number of melanomas detected was 1160, and non-melanoma lesions were 33,010. The performance of market-approved technology and clinician performance for classifying melanoma was highly heterogeneous, with sensitivity ranging from 16.4 to 100.0%, specificity between 40.0 and 98.3% and accuracy between 44.0 and 92.0%. Less heterogeneity was observed when clinicians worked in unison with AI, with sensitivity ranging between 83.3 and 100.0%, specificity between 83.7 and 87.3%, and accuracy between 86.4 and 86.9%. Conclusion: Instead of focusing on the performance of AI versus clinicians for classifying melanoma, more consistent performance has been obtained when clinicians’ work is supported by AI, facilitating management decisions and improving health outcomes.

## 1. Introduction

The annual global incidence of malignant melanoma (MM) has been reported at 325,000 cases, with associated mortality at 57,000 cases per annum [1]. The current rate of MM is projected to rise by a further 50% by 2040, with the mortality rate increasing to 96,000 annual cases [1]. Australia and New Zealand remain the two nations with the highest incidence of MM worldwide [2]. Major risk factors for melanoma include high nevi count, fairer skin, cumulative or high intermittent exposure to ultraviolet radiation, gene mutations of cyclin-dependent kinase inhibitor 2A (CDKN2A) and a personal or familial history of MM [3]. Advanced stages of melanoma continue to be associated with a poor prognosis, highlighting the importance of timely detection for a favourable patient outcome.

Patients screened during routine or opportunistic skin examinations are more likely to have an MM detected, coinciding with multiple skin biopsies, than non-screened individuals [4]. Although the MM rate continues to rise, this is not consistent with similar rates of reduced mortality [5], which may suggest overdiagnosis [6,7]. However, there is insufficient evidence linking overdiagnosis because of opportunistic screening [8]. This highlights the difficult task clinicians have of diagnosing MM early, particularly with feature-poor lesions that may have limited traits associated with invasive or metastatic malignancies. Dermatoscopy is one technique well accepted for improving the sensitivity of identifying malignant skin lesions compared to unaided visual inspection [9,10]. Additional technologies may further improve the detection of malignancies in the skin as well as a reduction in reliance on invasive procedures.

Artificial intelligence (AI) is proposed as a non-invasive tool that may help clinicians diagnose malignant lesions earlier and more accurately as well as limiting the necessity of cutaneous biopsies for otherwise benign lesions [11]. The patient perspective of AI in the diagnosis of MM has been reported to be favourable, particularly when proposed as an adjunct to a trained clinician [12]. The implementation of AI alongside dermatologists and specialists alike can aid in detecting malignancies that are hypothetically missed by human error [13].

The incorporation of computer-aided diagnosis to help detect skin cancer is not a new concept, with pilot studies examining the performance of AI for diagnosing MM published as early as 2005 [14]. It has been reported that by aiding in the early diagnosis of melanoma skin cancer, AI can reduce the associated morbidity and, ultimately, the mortality rate [15]. However, the emergence of improved machine learning technologies like convolutional neural networks (CNNs) with ‘dermatologist-level performance’ within the last decade has risen to prominence [16]. Many emerging studies remain in the realm of computer science, testing prospective algorithms on pre-build dermatoscopic image datasets including the ‘human against machine’ with 10,000 training images (HAM10000) dataset [17], international skin imaging collaboration (ISIC) challenges [18] and hospital Pedro Hispano (PH^2^) dataset [19]. Though studies report favourable AI performance, often surpassing specialists in cutaneous malignancy detection [20,21], pre-built databases represent an artificial scenario, and this may not replicate real-world settings that a clinician is likely to experience in general practice. 

Beyond melanoma classification, CNN-based algorithms have been implemented in different areas of medicine, including histopathology, medical photography and radiology [22]. Other medical imaging settings include diagnosing pneumonia via assessment of chest X-ray datasets [23] and visualization of internal organs to aid in disease identification [24].

Previous systematic reviews [25,26,27] have noted the scarcity of clinical studies, observing that most of the existing research is grounded in computer science, which may not represent real-world scenarios. In 2018, Chuchu and colleagues recorded two commercially available mobile applications that could identify MM from other suspicious-looking lesions [26]. More recently, Jones et al. identified two clinical studies from either primary care or community centres that investigated computer-aided detection of lesions suspected of cutaneous malignancies [27]. A common theme in melanoma-related AI detection is to classify suspect lesions as either malignant or benign. However, malignant lesions may refer to other cutaneous pathologies (keratinocyte carcinomas, melanoma, pigmented nevi, etc.), which makes determining AI’s performance for suspicion of MM difficult. Therefore, the aim of this systematic review was to compare studies that reported performance metrics when tasked with identifying MM from mobile applications or bedside CNNs that are commercially available.

## 2. Materials and Methods

### 2.1. Eligibility Criteria

This review follows the methodology as outlined by the preferred reporting items for systematic reviews and meta-analyses (PRISMA) [28]. The study was registered in PROSPERO on 30 November 2023 (Registration number: CRD42023484501). This review follows the PICO format where the population is a clinical population with melanoma, the intervention is artificial intelligence, the control is related to dermatoscopy or skin-related clinics and the outcome is diagnostic accuracy. Studies met the inclusion criteria if they were peer-reviewed, published papers between 2018 and 2023 and were reported in English. This time frame was chosen to limit repeating the work of Dick et al. [25], who previously published a meta-analysis of computer-aided diagnosis and melanoma in 2019.

Each paper was monitored for market-approved artificial intelligence available for clinical use during screening for eligibility. When studies met these criteria, the dataset of interest was assessed to determine the types of malignancies reported. Studies that reported cutaneous malignancies (melanoma in combination with other malignancies, i.e., pigmented basal cell carcinoma) but did not report the performance of melanoma separately were not eligible and were excluded. Studies that described building an algorithm for testing on prefabricated databases or ‘challenges’ or were limited to the field of computer science were excluded from eligibility. A full breakdown of the keywords used in each electronic database search is included in Appendix A.

### 2.2. Search Strategy

Five electronic databases (CINAHL, Medline, Scopus, ScienceDirect and Web of Science) were searched between 8 November 2023 and 31 December 2023, inclusive. Search terms were used that focused on the keywords “melanoma”, “performance metrics”, “artificial intelligence”, “detection” and “clinical settings”. Keywords were entered into three additional electronic databases (PEDro, Ovid and SPORTDiscus) but captured no articles, so the databases were not included in the final search.

Systematic review-type studies were not eligible for inclusion in the final study count. However, they were included in the search strategy as a means of checking the appropriateness of the keywords selected. Model papers were searched by title in the electronic databases selected to assess their availability (i.e., paper searched by title in SCOPUS). Then, the selected keywords were entered into the electronic databases. The selected search was then screened to identify if the keywords captured the model papers.

### 2.3. Reference Software and Study Selection

The reference management software EndNote (Version x9.3.3., Clarivate Analytics Philadelphia, PA, USA) was used to store the captured articles from the databases. After the records were obtained, duplicates were removed and the remaining articles were screened by title and abstract to assess potential eligibility. At this stage, any systematic reviews were also excluded. Full-text articles were then screened to assess eligibility. This screening process was completed by the lead author in consultation with two senior investigators.

### 2.4. Data Extraction and Data Synthesis

Data extraction was completed by the lead author in consultation with two senior investigators. Each study was screened for lesions under suspicion of MM. For MM to be included in this review, the ground truth of each lesion of suspicion was determined by histopathology. Performance metrics of interest were pre-determined by four investigators at the conception of this systematic review.

To determine the performance metrics, whether that be clinician or AI, four primary outcome variables of interest, true positive (TP), true negative (TN), false positive (FP) and false negative (FN), were selected. Both TP and TN dictate the ground truth, where a suspicious lesion with confirmed malignancy (via histopathology) is a TP and a queried lesion to out rule malignancy (a non-malignant lesion) is confirmed a TN. Values that are FP or FN are incorrect determinates, with an FN considered non-malignant and is diagnosed as malignant following histopathology and an FP originally considered malignant before a biopsy sent for laboratory analysis determines an alternative diagnosis. These values were recorded in the results from each study when included as outcome measures in the studies.

Performance metrics of interest also included sensitivity (Equation (1)), specificity (Equation (2)), accuracy (Equation (3)) and area under the curve for receiver operation characteristic (AUROC). Sensitivity is considered the performance of an analyst correctly identifying disease in the population of disease. Specificity measures the performance of determining the population without the disease. Finally, the accuracy is the analyst’s correct determination of the population with and without disease out of the entire population of interest (Figure 1). The equations used to calculate these performance metrics were as follows:Sensitivity = TP/(TP + FN)(1)
Specificity = TN/(FP + TN)(2)
Accuracy = (TP + TN)/(TP + FN + FP + TN)(3)

The performance metrics were recorded from each study based upon values relevant to the analysis of melanoma. When these performance metrics were not reported, these values were calculated based on the data provided in each study (where possible) and using Equations (1) through (3).

When reported, AUROC was included as an outcome variable of interest. Values of 0.5 are the minimum threshold and are recognized as happening by chance, values above 0.8 are considered clinically acceptable and values between 0.9 and 1.0 are highly accurate [29,30]. The reported performance metrics of each study were independently reviewed by two of the authors of this review. Any differences of opinion were resolved via consultation with senior investigators.

The performance metrics of the studies included in this review were summarized in a table format. Tables were categorized based on the technology (mobile applications, CNN or total body photography) or clinician. For example, the studies that investigated mobile applications were categorized together. If a study investigated multiple technologies (i.e., clinician, mobile application, CNN and/or total body photography), separate tables were created for each technology to help with readability. A narrative summary was provided alongside the tables to provide a summary of the sensitivity, specificity, accuracy and AUROC reported (or calculated if not provided) from the studies.

### 2.5. Critical Appraisal

An AXIS critical appraisal tool [31] between two authors was completed (independently) to assess the quality of the studies included in the review. The AXIS score was converted into percentages, with a score of 75% or above considered ‘good’, a score between 74.5% and 55.0% considered ‘fair’ and scores equal to or below 54.9% considered poor quality. Level of evidence was determined by guidelines as outlined by the National Health and Medical Research Council [32]. Interrater reliability was analysed with the statistics package SPSS (version 29.0 IBM SPSS Statistics for Windows, IBM Corp., Armonk, NY, USA), using Cohen’s Kappa Coefficient (k) [33].

## 3. Results

A total of 2772 studies were identified in electronic database searches. Following duplicate removal, screening of the captured literature and checking each paper for eligibility, 16 studies were included in the review (Figure 2).

### 3.1. Study Characteristics

There was a total of 1160 MM and 33,010 benign lesions in the 16 studies which met the inclusion criteria. Of the sixteen studies included in this review, eleven investigated bedside CNN performance, eight investigated CNNs’ performance versus clinicians, three compared CNNs alongside clinicians, three studies looked at mobile applications’ performances, three studies investigated 3D total body photography (TBP) and two studies reported performance metrics for 2D TBP (Figure 3).

The studies varied in geographic location, with Germany conducting a total of five published studies, four nations (Australia, Switzerland, UK, USA) having *n* = 2 studies and three nations (Canada, Romania and Netherlands combined, Spain) having *n* = 1 study each (Figure 4; top). The number of publications steadily increased from two in 2019, 2020 and 2021 to seven in 2023 (Figure 4; bottom).

### 3.2. Quality of Studies

Of the 16 articles included in this systematic review, 15 were rated as ‘good’ quality and one was rated as ‘fair’ quality between two authors independently. The average score between the articles was 83.4% (SD ± 5.9). The Cohen’s Kappa identified a ‘substantial agreement’ (k = 0.977, *p* < 0.001) between the two raters. A breakdown of the critical appraisal for each study criteria met for each study is included in Appendix A.

### 3.3. Assessment of Performance Metrics, Mobile Applications

Three articles [34,35,36] reported performance metrics from mobile applications. Subjects with MM ranged from *n* = 5 to 138 participants, and other diagnosis ranged from *n* = 55 to 6000 participants. With regard to performance metrics, the sensitivity ranged from 80.0 to 92.8%, the specificity was 60.0 to 95.0% and the accuracy was reported as 62.3 to 92.0%. One study [34] reported an AUROC value of 0.717 (Table 1).

### 3.4. Assessment of Performance Metrics, 3D TBP

Three articles [35,37,38] reported 3D TBP in clinical settings. Two of these studies [35,37] had small participant numbers, ranging from *n* = 6 to 10 participants with MM and *n* = 55 to 65 participants with an otherwise benign diagnosis. The sensitivity between the two studies was high and ranged from 83.3 to 90.0%, the specificity was lower at 63.6 to 64.6% and the accuracy of all histopathology tested lesions was 65.6 to 68.0%. Of these two studies, one study [37] reported an AUROC value of 0.92. One study [38] followed a different methodology and did not report values for sensitivity, specificity and accuracy. However, the AUROC was reported (0.9399) and was included (Table 2).

### 3.5. Assessment of Performance Metrics, 2D TBP

Two articles [35,37] reported 2D TBP on patients. Between the two studies, there were small participant numbers, ranging from *n* = 6 to 10 participants with MM and *n* = 55 to 60 participants with non-malignant diagnosis. The sensitivity ranged from 70.0 to 83.3%, the specificity for both studies was 40.0% and the reported accuracy was 44.0 to 44.3%. Of these two studies, one study [37] reported an AUROC value of 0.68% (Table 3).

### 3.6. Assessment of Performance Metrics, CNN with Clinician

Three articles [35,37,39] compared clinicians and AI in unison. For these studies, MM diagnosed ranged from *n* = 6 to 38 participants and *n* = 55 to 190 participants had otherwise benign diagnosis. The sensitivity was good and ranged from 83.3 to 100.0%, the specificity was between 83.7 and 87.3% and the accuracy of diagnosis was 86.4 and 86.9%. Two of these studies [37,39] reported AUC values of 0.88 to 0.968 (Table 4).

### 3.7. Assessment of Performance Metrics, CNN

There were 11 articles [39,40,41,42,43,44,45,46,47,48,49] which reported performance on bedside CNNs. One article’s [42] main outcome was monitoring changes in sequential digital dermatoscopy and presented results differently and could not be compared. For the other studies, MM ranged between *n* = 15 and 140 participants and others (lesions that were not melanoma) ranged from *n* = 33 to 4495 participants. The sensitivity had a wide range from 16.4 to 100.0%, the specificity ranged from 54.4 to 98.3% and the accuracy ranged from 54.2 to 87.7%. Six of the eleven studies [39,40,41,45,47,48] reported AUROC values, which ranged from 0.54 to 0.969 (Table 5).

### 3.8. Assessment of Performance Metrics, CNN versus Clinician

Eight studies [34,35,37,39,43,44,46,48] reported clinician metrics. Studies reporting MM had *n* = 6 to 124 participants and others (lesions that were not melanoma) had *n* = 36 to 190 participants. The sensitivity ranged from 41.8% (novice practitioners) to 96.6% (dermatologists). Of the remaining performance metrics for clinicians, the specificity ranged from 32.2 to 92.7% and the accuracy was reported to be 52.0 to 92.0%. Three of these studies [37,39,48] reported clinician values for AUROC, which ranged from 0.778 to 0.91 (Table 6). When comparing clinicians with CNNs, the sensitivity (41.8 to 96.6% vs. 16.4 to 100.0%), specificity (32.2 to 92.7% vs. 54.4 to 98.3%), accuracy (52.0 to 92.0% vs. 54.2 to 87.7%) and AUROC (0.778 to 0.91 vs. 0.54 to 0.969) were heterogeneous.

### 3.9. Summary of Performance Metrics

Of the captured studies of this systematic review, performance metrics were reported for mobile phone-based applications, TBP (both 2D and 3D variants), bedside CNNs as standalone devices, clinicians in comparison with AI and clinicians in unison with AI. A summary of the reported performance metrics from the various AI technologies and clinicians alike is included (Table 7).

## 4. Discussion

This systematic review compared studies published in the last six years that reported performance metrics when tasked with classifying melanoma from mobile applications or commercial bedside CNNs. The studies captured in this systematic review highlight a high degree of variability in methodology, technology used, geographic locations and subject populations. As such, it is unsurprising to see a wide degree of promising and questionable performances when detecting melanoma from non-malignant pathologies.

This systematic review reaffirms previous reviews [25,27] that the majority of studies investigating AI’s performance remain in the space of computer science, with relatively few captured studies identified in the field of medicine where the technology was used on actual patients. Studies in computer science often reported outstanding performance (i.e., high sensitivity, specificity and accuracy) when the computer-aided diagnosis is tested on pre-built image datasets. However, the performance of these algorithms may not replicate real-world situations experienced in clinical practice.

It should be noted that at the present time AI does not replace a clinician’s diagnostic acumen for detecting keratinocyte carcinomas, melanoma or any skin-related pathologies. Where AI may position itself in current clinical practice is by monitoring and detecting early changes occurring in cutaneous lesions and alerting trained clinicians to detect lesions that may warrant heightened scrutiny. Nevertheless, testing of these market-available units in a clinical setting is vital if the software is to improve to the point of being a useful (i.e., valid) tool for experienced dermatologists or general practitioners (GPs) who have a focus on skin-related diagnosis.

### 4.1. Sensitivity and Specificity: Which Is More Important?

A low sensitivity indicates a greater likelihood of missing patients with the disease, in this instance, melanoma. A low value in sensitivity is an increased chance of false negative findings. For melanoma, this may result in a significant impact on survivability, with advanced stages of disease remaining associated with poorer prognosis and poor patient outcomes [50]. Low specificity values will also result in more patients likely to be treated with the disease unnecessarily. This would result in overtreatment, where patients undergo unnecessary medical interventions such as excisional biopsies. The take-home message is that neither sensitivity nor specificity should be considered without consideration of both metrics.

The position statement of the European Academy of Dermatology and Venerology (EADV) has avoided stating the minimally acceptable accuracy of mobile applications and web-based services for skin diseases, citing risk–benefit reasons [51]. Though dictating a minimum figure on the performance of AI for melanoma classification is tempting, our group agrees that this may present more harm than good. The use of AI relating to melanoma detection does not yet replace the gold standard of diagnosis, which remains by biopsy of suspect moles or lesions for determination via histopathology [52].

### 4.2. Mobile Applications and Their Performance

There were few studies identified that investigated mobile applications’ performance metrics to correctly classify melanoma. Of the three studies captured in this review, one study [34] compared a market-available mobile application against clinicians on an open access challenge dataset (M-Class) for melanoma detection. Previously, Brinker and colleagues tested this same dataset against 157 dermatologists [53], with improved sensitivity (80.0% to 89.4%) but lower specificity (95.0% to 64.4%). The performance metrics of the mobile application on the open challenge dataset were greater for sensitivity, specificity and accuracy compared to the two studies [35,36] that investigated patient data in clinics. This would suggest that the performance of mobile applications for detecting melanoma in artificial situations, such as challenge datasets, may not replicate the experiences of these applications in real world settings.

Consumer support for the use of mobile applications for teledermoscopy has been overwhelmingly supportive [54]. While not a replacement for a traditional skin cancer screening consultation with an experienced clinician, favourable accuracy from the three studies suggests that mobile applications may identify lesions of suspicion more often than not. This would be particularly useful for persons in rural and remote areas without easy or regular access to specialists in skin cancer detection.

### 4.3. Total Body Photography

Hornung and colleagues have previously investigated TBP performance, noting greater detection of in situ MM in studies, as well as fewer invasive melanomas [55]. The systematic review by Hornung and colleagues reflects the studies captured in this systematic review, highlighted by favourable sensitivity performance, albeit decreased specificity. Low specificity indicates greater detection of subjects without a disease, treated like they have the disease (higher false positive values). Low specificity values result in overdiagnosis, a topic of discussion for melanoma rates rising dramatically without the same rate of increase in mortality [6,56].

### 4.4. Clinicians and AI, Working in Unison

Studies investigating clinicians and AI working in unison were less variable than comparing clinicians versus AI alone. Though limited to three published studies [35,37,39], the performance metrics are promising, with much-improved specificity values (83.7 to 87.3%) alongside sensitivity (83.3 to 100.0%) performance. Jain et al. [57] retrospectively showed that AI has the potential to improve primary care physicians and nurse practitioner’s ability to correctly diagnose skin conditions, reducing the reliance on surgical intervention on benign lesions. Tschandl and colleagues [58] also noted improvements in clinical decision-making over humans or machines working independently. This is a recent shift in approach, with many studies previously focused on the comparison between clinician performance and ‘machine’ performance.

### 4.5. Performance of Bedside CNN versus Clinicians

The studies that compared both clinician and CNN performance metrics [34,35,37,39,43,44,46,48] showed high heterogeneity. No clear analyst (either clinician or CNN) could clearly be determined as more effective at correctly classifying MM in the subjects of these studies. There was a high degree of variance between studies, which may be a result of a multitude of factors, including the number of participants, the demographics (not always included in the studies), the type of CNN compared (mobile, bedside and TBP) and the type of MM detected (number of in situ vs. invasive). This may be, in part, a measure of the variance of geographic locations of these studies, with six countries (Australia [46], Canada [44], Germany [39,43], Switzerland [37], UK [48,49] and USA [34]) recording studies.

By location, the two studies [46,47] published by Australian authors reported comparatively lower sensitivity values in CNNs than previously reported [39,40,41,43,44,45,48,49]. Australia remains the nation with the highest incidence of melanoma [2]. As such, early diagnosis of MM is paramount. It has been reported that Australian GPs diagnose more than 75% of all melanomas within the primary care settings, with the majority of these in situ MM [59]. This may be partly attributed to childhood and chronic ultraviolet radiation of high intensity in Australia (and, by geographic extension and similarly high incidence, New Zealand) with observable skin damage reported decades earlier compared to other fair-skinned nations like Canada, the UK and the USA [60]. The authors hypothesize that Australian GPs in primary care skin-clinics, with more than five years’ experience diagnosing skin cancer, are likely to detect suspect lesions in early stages, often with only subtle clues to malignancy. The authors propose that an AI architecture may benefit from images with early evolving melanoma, which may be more culturally relevant to nations such as Australia and New Zealand which have a high incidence of melanoma. 

Novice family practitioners or clinicians with limited training and experience in identifying MM may benefit most from CNNs. In general, less-experienced clinicians in melanoma detection were less accurate than more experienced or specialist practitioners [61].

### 4.6. Future Directions

Market-approved CNNs are provided with an input in a 2D plane and are tasked with providing an output that highlights the likelihood of a lesion being suspicious to malignancy or not. This contrasts with clinicians, who are provided with a patient’s previous and family history of skin cancer and the ability to identify if a lesion is raised or not. Interpretation of colours via a dermatoscope can help a clinician identify the presence of melanin pigment at depths in the skin not usually encountered, which may suggest invasive progression of malignancy [62]. Bedside CNN performance, followed by assessing tissue analysis via histopathology, may be one avenue of investigating malignancy characteristics and the performance of market-approved AI.

Other options of AI detecting changes is by sequential digital dermatoscopy or the monitoring of several photographs of a lesion over time to help identify changes. In clinical practice, Winkler and colleagues [42] surmised that as of 2022, CNNs cannot replace serial monitoring and that there was an inherent need of CNNs that integrate information as part of this analysis. In 2019, Polap [63] proposed an option of CNNs capable of integrating medical data as part of skin cancer classification. The performance of this modified CNN was promising, with correct classification of MM at 89.5%. However, this analysis was performed on the pre-determined databases HAM10000 [17] and ISIC2017 [64]. This systematic review also identified studies investigating AI algorithms for skin cancer identification that are still not market-approved and therefore were not selected. However, future research within clinical settings is needed to bring novel advances in AI technology and the application of new AI algorithms.

### 4.7. Limitations

This systematic review has limitations. The data obtained from the 16 studies of this review were highly varied. This is most likely due to the differences in study methodology and varied number and geographic location of the subjects, as well as the limited market-approved CNNs available. Of the few studies available looking at market available CNNs, 15 studies [65,66,67,68,69,70,71,72,73,74,75,76,77,78,79] classified lesions as malignant (melanoma in combination with other cutaneous carcinomas) versus benign which could not be included in this review. This systematic review focused on the melanoma class of cutaneous cancer. However, this means that half of the market available studies were excluded in this review. With the increasing trend of studies reporting AI performance metrics when classifying melanoma in the field of medicine, it is suggested that a review of the literature in years to come is warranted in the form of a meta-analysis. Though not limited to this systematic review, there remains an inherent lack of testing on darker-skinned individuals, particularly Fitzpatrick skin types between four to six.

The methodology of this review did not include databases from the grey literature, such as Google Scholar or from theses. In addition, not including trial registries may be considered a limitation. However, trial registries focus on the design of the proposed study and less on the performance outcomes of extended clinical results, which is not the focus on this systematic review. 

Of the market-approved CNNs available, there was limited transparency available for the architecture used for all studies. From the studies investigated, both ‘FotoFinder’ and ‘quantusSkin’ have reported using CNN variants based on GoogLeNet Inception algorithms [39,40,41,42,44,45,47]. Upon request, Canfield [35,37,38] shared they utilize two AI applications, one based on the EfficientNetV2 architecture and an additional custom CNN. Our group had reached out to the technologies captured in our review (i.e., MetaOptima, SkinAnalytics, SkinVision and Triage Technologies) to better understand the various architectures available but at the time of publication did not receive a reply. Though this is disappointing, we can appreciate a company’s right not to disclose their technology for a perceived (or real) commercial advantage and to the greater population for risk of this technology being replicated and bypassing their licensing terms. However, this lack of disclosure does highlight a limitation from a research perspective if CNN variants are to be compared.

## 5. Conclusions

The implementation of AI in clinical practice to assist in the early detection of melanoma remains promising, particularly with the prospect of improved diagnostic accuracy. However, the heterogeneity of studies reporting on AI performance for classifying melanoma in clinical practice indicates it currently cannot be recommended as a reliable tool for clinicians. For predictive algorithms to improve, high-quality images that capture malignant lesions are necessary to help strengthen the architecture that makes up market-approved CNNs. It may be necessary that this architecture includes a broader variety of skin types and degrees of sun-damaged skin. This is highly relevant to countries with high incidences of melanoma to aid in early detection and more favourable prognosis. Most promisingly, the collaboration of clinicians and AI-powered machines may be a shift away from comparing clinician’s performance against that of AI. Such a partnership may result in enhanced outcomes for the management and timely detection of malignant lesions, particularly melanoma.

## Figures and Tables

**Figure 1 cancers-16-01443-f001:**
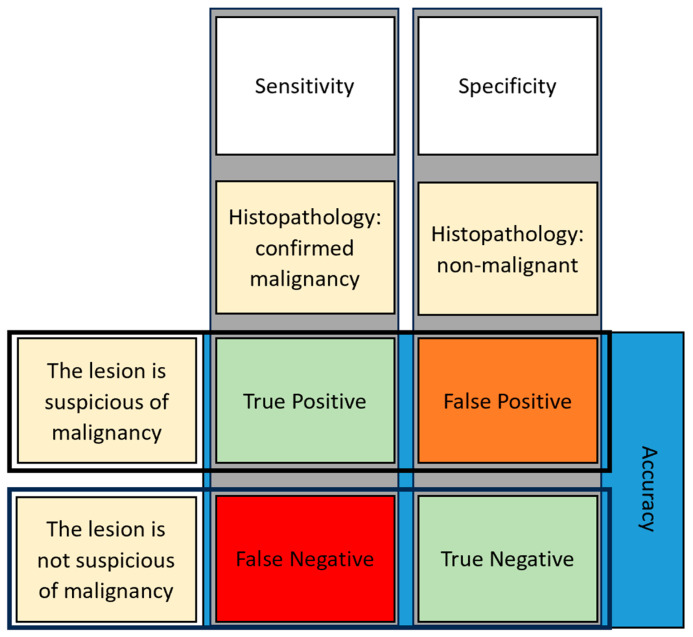
A visual representation of sensitivity and specificity.

**Figure 2 cancers-16-01443-f002:**
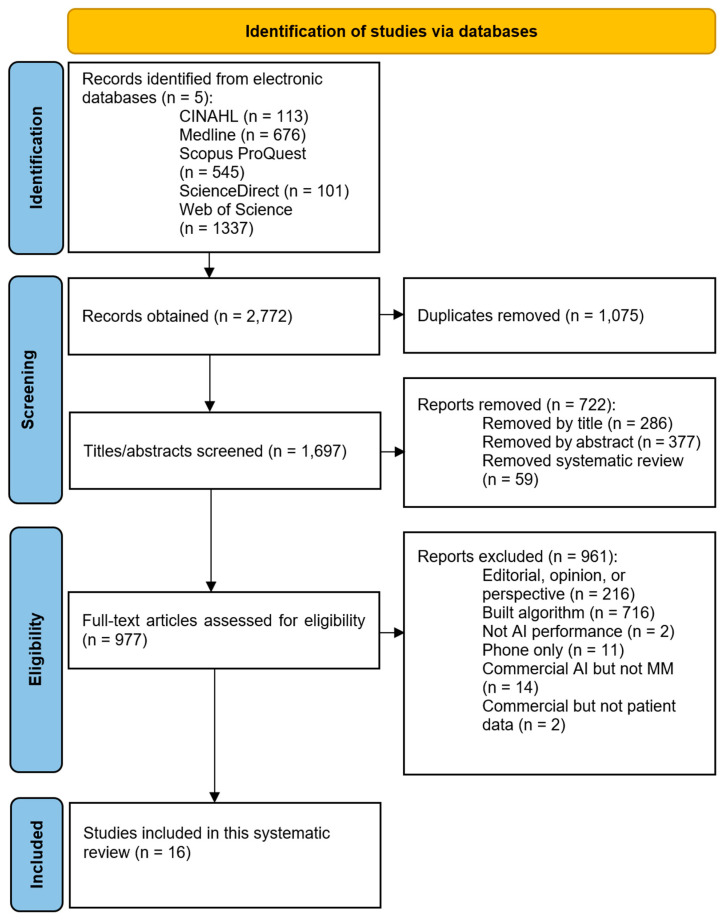
PRISMA flow diagram.

**Figure 3 cancers-16-01443-f003:**
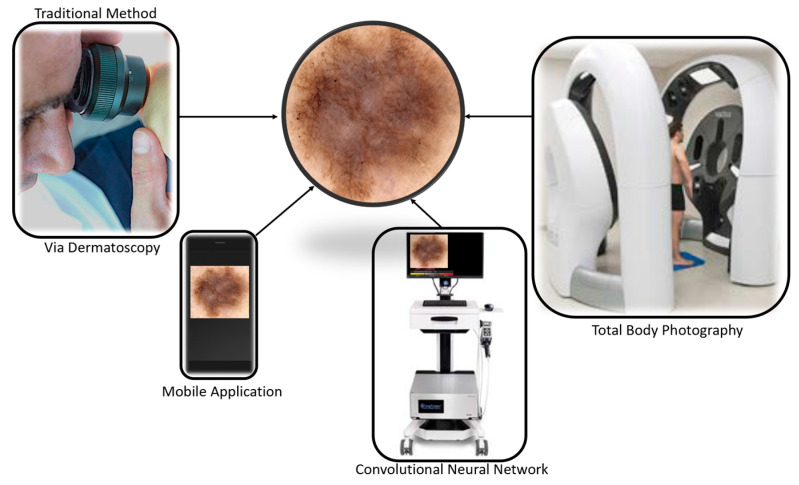
Detection techniques captured in this systematic review. Traditional methods involve either opportunistic or routine skin examination via dermatoscopy. Emerging technologies include mobile applications, bedside convolutional neural networks and serial monitoring with the aid of total body photography.

**Figure 4 cancers-16-01443-f004:**
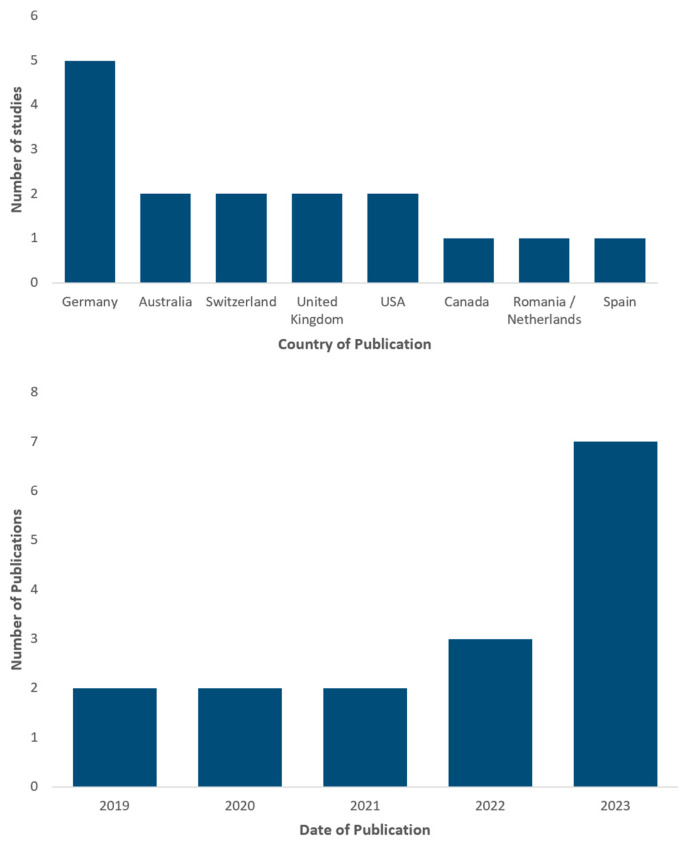
(**top**) Describes the country location of publication; (**bottom**) describes the year the included studies were published.

**Table 1 cancers-16-01443-t001:** Performance metrics of melanoma classification, mobile applications.

Author, Year, Country	Patients	Technology	True Positive	False Negative	False Positive	True Negative	Sensitivity [95% CI] *	Specificity [95% CI] *	Accuracy [95% CI] *	AUROC [95% CI] *
Anderson et al., 2023, USA [34]	MM (*n* = 20), Other (*n* = 80	CNN (triage; mobile application)	16	4	4	76	80.0	95.0	92.0	
Jahn et al., 2022, Switzerland [35]	MM (*n* = 6), Other (*n* = 55)	CNN (SkinVision; mobile application)	5	1	22	33	83.3	60.0	62.3	0.717
Udrea et al., 2020, Romania/Netherlands [36]	MM (*n* = 138), Other (*n* = 6000)	CNN SkinVision; mobile application)	128	10	1302	4698	92.8 [87.8–96.5]	78.3 [77.2–79.3]	78.6	

where * 95% CI = 95% confidence interval; included in table when study reported it in results, AI = Artificial intelligence, AUROC = Area under the curve receiver operative characteristic, CNN = Convolutional neural network.

**Table 2 cancers-16-01443-t002:** Performance metrics of melanoma classification, 3D TBP.

Author, Year, Country	Patients	Technology	True Positive	False Negative	False Positive	True Negative	Sensitivity [95% CI] *	Specificity [95% CI] *	Accuracy [95% CI] *	AUROC [95% CI] *
Cerminara et al., 2023, Switzerland [37]	MM (*n* = 10), Other (*n* = 65)	CNN (Canfield; 3D Vectra WB360)	9	1	23	42	90.0	64.6	68.0	0.92 [0.85–1.00]
Jahn et al., 2022, Switzerland [35]	MM (*n* = 6), Other (*n* = 55)	CNN (Canfield; 3D Vectra WB360)	5	1	20	35	83.3	63.6	65.6	
Marchetti et al., 2023, USA [38]	MM (*n* = 43), Other (*n* = 22,489)	CNN (Canfield; 3D Vectra WB360)								0.9399 [0.92–0.96]

where * 95% CI = 95% confidence interval; included in table when study reported it in results, AI = Artificial intelligence, AUROC = Area under the curve receiver operative characteristic, CNN = Convolutional neural network, TBP = Total body photography.

**Table 3 cancers-16-01443-t003:** Performance metrics of melanoma classification; 2D TBP.

Author, Year, Country	Patients	Technology	True Positive	False Negative	False Positive	True Negative	Sensitivity [95% CI] *	Specificity [95% CI] *	Accuracy [95% CI] *	AUROC [95% CI] *
Cerminara et al., 2023, Switzerland [37]	MM (*n* = 10), Other (*n* = 65)	CNN (FotoFinder; 2D TBP)	7	3	39	26	70.0	40.0	44.0	0.68 [0.46–0.90]
Jahn et al., 2022, Switzerland [35]	MM (*n* = 6), Other (*n* = 55)	CNN (FotoFinder; 2D TBP)	5	1	33	22	83.3	40.0	44.3	

Where: * 95% CI = 95% confidence interval; included in table when study reported it in results, AI = Artificial intelligence, AUROC = Area under the curve receiver operative characteristic, CNN = Convolutional neural network, TBP = Total body photography.

**Table 4 cancers-16-01443-t004:** Performance metrics of melanoma classification, clinician in unison with AI.

Author, Year, Country	Patients	Clinician or Clinician with AI	True Positive	False Negative	False Positive	True Negative	Sensitivity [95% CI] *	Specificity [95% CI] *	Accuracy [95% CI] *	AUROC [95% CI] *
Cerminara et al., 2023, Switzerland [37]	MM (*n* = 10), Other (*n* = 65)	Dermatologist plus AI	9	1	9	56	90.0	86.2	86.7	0.88 [0.80–1.00]
Jahn et al., 2022, Switzerland [35]	MM (*n* = 6), Other (*n* = 55)	Dermatologist (average) plus AI	5	1	7	48	83.3	87.3	86.9	
		*Beginner (<2 years) #*	*4*	*1*	*6*	*34*	*80.0*	*85.0*	*84.4*	
		*Skilled (2–5 years) #*	*0*	*0*	*1*	*4*	*-*	*80.0*	*80.0*	
		*Expert (>5 years) #*	*1*	*0*	*0*	*10*	*100.0*	*100.0*	*100.0*	
Winkler et al., 2023, Germany [39]	MM (*n* = 38), Other (*n* = 190)	Dermatologist plus AI	38	0	31	159	100.0 [90.8–100]	83.7 [77.8–88.3]	86.4 [81.3–90.3]	0.968

where * 95% CI = 95% confidence interval; included in table when study reported it in results, # italic formatting to indicate Dermatologist subgroups, AI = Artificial intelligence, AUROC = Area under the curve receiver operative characteristic, CNN = Convolutional neural network, TBP = Total body photography.

**Table 5 cancers-16-01443-t005:** Performance metrics of melanoma classification, CNN.

Author, Year, Country	Patients	Technology	True Positive	False Negative	False Positive	True Negative	Sensitivity [95% CI] *	Specificity [95% CI] *	Accuracy [95% CI] *	AUROC [95% CI] *
Phillips et al., 2019, United Kingdom [48]	MM (*n* = 79), Other (*n* = 310)	CNN (SkinAnalytics; DERM with iPhone)	62	17			78.5			0.879
	MM (*n* = 76), Other (*n* = 300)	CNN (SkinAnalytics; DERM with Galaxy5)	54	22			71.1			0.823
	MM (*n* = 51), Other (*n* = 220)	CNN (SkinAnalytics; DERM with DSLR)	38	13			74.5			0.850
Thomas et al., 2023, United Kingdom [49]	MM (*n* = 140), Other (*n* = 4495)	CNN (SkinAnalytics; DERMvA-UHB)	133	7	1852	2643	95.0 [90.0–97.6]	58.8 [57.4–60.2]	59.9	
	MM (*n* = 33), Other (*n* = 676)	CNN (SkinAnalytics; DERMvA-WSFT)	32	33	249	427	97.0 [84.7–99.5]	63.2 [59.5–66.7]	64.7	
	MM (*n* = 58), Other (*n* = 2527)	CNN (SkinAnalytics; DERMvB-UHB)	58	0	482	2045	100.0 [93.8–100]	80.9 [79.3–82.4]	81.4	
	MM (*n* = 18), Other (*n* = 624)	CNN (SkinAnalytics; DERMvB-WSFT)	18	0	122	502	100.0 [82.4–100]	80.4 [77.2–83.4]	81.0	
Fink et al., 2020, Germany [43]	MM (*n* = 36), Other (*n* = 36)	CNN (Fotofinder; MoleAnalyzer Pro)	35	1	8	28	97.1 [82.7–99.6]	78.8 [62.8–89.1]	87.5	
MacLellan et al., 2021, Canada [44]	MM (*n* = 59), Other (*n* = 150)	CNN (FotoFinder; MoleAnalyzer Pro)	52	7	32	118	88.1 [79.4–96.9]	78.8 [71.5–86.2]	81.3	
	MM (*n* = 59), Other (*n* = 150)	CNN2 (FotoFinder; MoleAnalyzer Tuebinger)	49	10	37	113	83.1 [72.6–93.6]	75.2 [67.3–83.1]	77.5	
Miller et al., 2023, Australia [47]	MM (*n* = 15), Other (*n* = 33)	CNN (FotoFinder; MoleAnalyzer Pro)	8	7	15	33	53.3	54.4	54.2	0.540
Winkler et al., 2023, Germany [39]	MM (*n* = 38), Other (*n* = 190)	CNN (FotoFinder; MoleAnalyzer Pro)	31	7	21	169	81.6 [66.6–90.8]	88.9 [77.8–88.3]	87.7 [82.8–91.4]	0.904
Winkler et al., 2019, Germany [40]	MM (*n* = 23), Other (*n* = 107)	CNN (FotoFinder; MoleAnalyzer Pro)	22	1	17	90	95.7 [79.0–99.2]	84.1 [76.0–89.8]	86.2	0.969
Winkler et al., 2021, Germany [41]	MM (*n* = 23), Other (*n* = 107)	CNN (FotoFinder; MoleAnalyzer Pro)	20	3	13	94	87.0 [67.9–95.5]	87.9 [80.3–92.8]	87.7	0.953 [0.914–0.992]
Winkler et al., 2022, Germany [42]	MM (*n* = 59), Other (*n* = 236)	CNN (FotoFinder; MoleAnalyzer Pro)								
Menzies et al., 2023, Australia [46]	MM (*n* = 55), Other (*n* = 117)	CNN (MetaOptima; 7-class)	28	27	7	110	50.9	94.0	80.2	
		CNN (MetaOptima; ISIC)	9	46	2	115	16.4	98.3	72.1	
Martin-Gonzalez et al., 2022, Spain [45]	MM (*n* = 55), Other (*n* = 177)	CNN (quantusSKIN)	38	17	34	142	69.1	80.2	77.6	0.802

where * 95% CI = 95% confidence interval; included in table when study reported it in results, AI = Artificial intelligence, AUROC = Area under the curve receiver operative characteristic, CNN = Convolutional neural network, TBP = Total body photography.

**Table 6 cancers-16-01443-t006:** Performance metrics of melanoma classification, clinician.

Author, year, country	Patients	Clinician or clinician with AI	True positive	False negative	False positive	True negative	Sensitivity [95% CI] *	Specificity [95% CI] *	Accuracy [95% CI] *	AUROC [95% CI] *
Anderson et al., 2023, USA [34]	MM (*n* = 20), Other (*n* = 80)	Primary care providers					67.0	48.0	52.0	
		*Family physicians #*					*78.0*	*41.0*	*48.0*	
		*Mid-level provider #*					*61.0*	*53.0*	*55.0*	
		Dermatologist					77.0	57.0	61.0	
Cerminara et al., 2023, Switzerland [37]	MM (*n* = 10), Other (*n* = 65)	Dermatologist	9	1	5	60	90.0	92.3	92.0	0.91 [0.80–1.00]
Fink et al., 2020, Germany [43]	MM (*n* = 36), Other (*n* = 36)	Dermatologist					90.6 [84.1–94.7]	71.0 [62.6–78.1]		
		*Beginner (<2 years) #*					*90.9* *[82.4–95.5]*	*55.1* *[45.7–64.2]*		
		*Skilled (2–5 years) #*					*93.3* *[86.3–96.9]*	*74.2* *[64.4–82.0]*		
		*Expert (>5 years) #*					*86.7* *[77.7–92.4]*	*80.6* *[70.2–88.0]*		
Jahn et al., 2022, Switzerland [35]	MM (*n* = 6), Other (*n* = 55)	Dermatologist (average)	5	1	4	51	83.3	92.7	91.8	
		*Beginner (<2 years) #*	*4*	*1*	*3*	*37*	*80.0*	*92.5*	*91.1*	
		*Skilled (2–5 years) #*	*0*	*0*	*1*	*4*	*-*	*80.0*	*80.0*	
		*Expert (>5 years) #*	*1*	*0*	*0*	*10*	*100.0*	*100.0*	*100*	
Maclellan et al., 2021, Canada [44]	MM (*n* = 59), Other (*n* = 150)	Dermatologist	57	2	44	106	96.6 [91.9–100.0]	32.2 [18.4–46.0]	78.0	
Menzies et al., 2023, Australia [46]	MM (*n* = 55), Other (*n* = 117)	Specialist	34	21	17	100	61.8	85.5	77.9	
		Novice	23	32	32	85	41.8	72.6	62.8	
Phillips et al., 2019, United Kingdom [48]	MM (*n* = 125), Other (*n* = 426)	Clinician	84	41			67.2			0.778
Winkler et al., 2023, Germany [39]	MM (*n* = 38), Other (*n* = 190)	Dermatologist	32	6	53	137	84.2 [69.9–92.6]	72.1 [65.3–78.0]	74.1 [68.1–79.4]	0.895

where * 95% CI = 95% confidence interval; included in table when study reported it in results, # italic formatting to indicate dermatologist or family physician subgroups, AI = Artificial intelligence, AUROC = Area under the curve receiver operative characteristic, CNN = Convolutional neural network, TBP = Total body photography.

**Table 7 cancers-16-01443-t007:** Summary of the available technologies.

Type of Technology (*n* = Number of Studies)	Performance Metric	Lower Limit %	Upper Limit %
Mobile Applications (*n* = 3) [34,35,36]	Sensitivity	80.0 [34]	92.8 [36]
	Specificity	60.0 [35]	95.0 [34]
	Accuracy	62.3 [35]	92.0 [34]
	AUROC	0.717 [35]	0.717 [35]
3D TBP (*n* = 3) [35,37,38]	Sensitivity	83.3 [35]	90.0 [37]
	Specificity	63.6 [35]	64.6 [37]
	Accuracy	65.6 [35]	68.0 [37]
	AUROC	0.92 [37]	0.94 [38]
2D TBP (*n* = 2) [35,37]	Sensitivity	70.0 [37]	83.3 [35]
	Specificity	40.0 [35,37]	40.0 [35,37]
	Accuracy	44.0 [37]	44.3 [35]
	AUROC	0.68 [37]	0.68 [37]
Clinicians in unison with AI (*n* = 3) [35,37,39]	Sensitivity	83.3 [35]	100.0 [39]
	Specificity	83.7 [39]	87.3 [35]
	Accuracy	86.4 [39]	86.9 [35]
	AUROC	0.88 [37]	0.968 [39]
CNN (*n* = 11) [39,40,41,42,43,44,45,46,47,48,49]	Sensitivity	16.4 [46]	100.0 [49]
	Specificity	54.4 [47]	98.3 [46]
	Accuracy	54.2 [47]	87.7 [39,41]
	AUROC	0.540 [47]	0.969 [40]
Clinician, No AI (*n* = 8) [34,35,37,39,43,44,46,48]	Sensitivity	41.8 [46]	96.6 [44]
	Specificity	32.2 [44]	92.7 [35]
	Accuracy	52.0 [34]	92.0 [37]
	AUROC	0.778 [48]	0.91 [37]

AI = Artificial intelligence, AUROC = Area under the curve receiver operative characteristic, CNN = Convolutional neural network, TBP = Total body photography.

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
