# Peer review of "Performance of Commercial Dermatoscopic Systems That Incorporate Artificial Intelligence for the Identification of Melanoma in General Practice: A Systematic Review"

_cancers, 2024, doi:10.3390/cancers16071443_

Round 1

Reviewer 1 Report

Comments and Suggestions for Authors

Good work. Congratulations.

Reviewer 2 Report

Comments and Suggestions for Authors

The systematic review explores and analyzes Artificial Intelligence approaches in melanoma detection. The paper is interesting, but i think that the paper still needs more improvements:

1) Based on your paper, CNN is the main tool. Can you examine other research to find out different approaches and discuss this issue?

2) What accuracy is necessary for the practical application of such AI methods? More information and some background would be good to discuss

3) In terms of CNN, there is no information, analysis or even comparison in terms of specific approaches. For instance, is it a basic CNN model? Or learning transfer? Last year brought an attention mechanism, so is it used in such problems?

4) Add some analysis in terms of charts and visualization to show the current state

5) Future directions and steps should be evaluated and discussed in detail to show the current trends and give the readers ideas about what next

6) Check such papers as: Analysis of skin marks through the use of intelligent things; where different CNN model was used. Such a solution could be discussed. 

7) AI methods analyze images mainly, but what about different shapes/colors etc. - which of these issues is the most challenging right now?

Reviewer 3 Report

Comments and Suggestions for Authors

The authors submitted a manuscript addressing the performance of commercial available diagnostic systems that are aided by artificial intelligence for identification of melanoma.

Given the high prevalence of early dectection of melanoma, the authors report on a very relevant topic. The introduction provides sufficient background information on melanoma and current use of AI. The chosen experimental techniques are appropriate. The presented results and the discussion are reported in a clear and interpretable manner.

The subsumption of this review appears to be well-structured and provides clear information. The only point I would recommend, is to correct a typing error in line 353: The author is called Brinker not Brinkler.

Reviewer 4 Report

Comments and Suggestions for Authors

The presented systematic review has a good level of organization and clarity in outlining the research. The structure effectively contextualizes the findings and their interpretation.

The results are predominantly presented through the lens of true positives (TP), true negatives (TN), false positives (FP), false negatives (FN), and the Area Under the Curve (AUC), as reported by the studies.  When possible, the authors computate accuracy, sensitivity, and specificity, adding depth to the analysis. However, it would enhance the discussion to introduce the f1-score, which provides a balanced measure of precision and sensitivity. Given the likelihood of unbalanced datasets for melanoma, the f1-score offers a more informative metric than accuracy alone and should be included in the analysis.

Some minor issues detected:  

- What where the criterion for the critical appraisal of included studies?

- Affirmation in line 259 seems to be contradictory with information in Table 1.  

- Line 267, first sentence - reference to the three mentioned articles is missing (only one provided)

Round 2

Reviewer 2 Report

Comments and Suggestions for Authors

The paper can be accepted